# Impact of Introversion-Extraversion Personality Traits on Knowledge-Sharing Intention in Online Health Communities: A Multi-Group Analysis

**Mian Lv [1], Yongbo Sun [2] and Binbin Shi [2,\*]**

[1]   School of Foreign Languages, Beijing Technology and Business University, Beijing 100048, China
[2]   Business School, Beijing Technology and Business University, Beijing 100048, China
\*   Correspondence: sb990826@163.com

**Abstract:** This paper investigates the influence of introversion-extraversion personality traits on the knowledge-sharing intention of online health communities (OHCs) using personality trait theory and social capital theory. This study investigates two types of users in OHCs—doctors and patients—and compares and analyses the knowledge-sharing paths of these two types of users. The results show that extraversion personality, interaction, and reciprocity positively influence the physicians' and patients' knowledge-sharing intention; for both types of users, interaction partially mediates between extraversion personality and knowledge-sharing intention, and reciprocity partially mediates between interaction and knowledge-sharing intention. Comparative analyses show that the physicians' introversion-extraversion personality traits have stronger positive effects on interaction, and interaction has stronger positive effects on trust and reciprocity than patients, the physicians' trust and reciprocity have stronger positive effects on knowledge-sharing intention than patients, and the physicians' introversion personality traits have stronger positive effects on knowledge-sharing intentions than patients. This study enriches the theory of user knowledge-sharing in OHCs while advancing the managers' understanding of what motivates users' knowledge-sharing intention.

**Keywords:** social capital theory; personality traits; knowledge-sharing; OHCs; comparative study

## 1. Introduction

Online health services are a type of service that is built on a telemedicine system in which doctors provide online services through the mobile Internet to achieve cross-territory diagnosis and treatment without the restrictions of time and place [1]. In the context of the novel coronavirus epidemic, many offline medical services find it difficult to operate normally, while there is a temporary shortage of medical resources in some areas where there is a severe epidemic, and online health communities (OHCs) have become an important way for patients to receive an initial consultation and relevant health information support [2]. Health information is considered the "key to quality care", and users' health information search behavior can help them make better care decisions and prevent unhealthy behavior promptly [3]. In recent years, with the development of social networks and increasing citizen health awareness, OHCs have played an important role in users' healthcare knowledge-seeking and disease management experience sharing [4]. For example, PatientsLikeMe, WebMD, and MedHelp provide an Internet-based platform that includes not only patients but also doctors [5], where community members can share health knowledge through interactive modules such as private messages or discussion forums. OHCs provide an open platform for users to access medical resources and share knowledge, experiences, and emotions, which provide effective social support for users, encourage them to prevent diseases in advance, and relieve patients' stress and anxiety [6]. Online health services can improve the quality of care by maintaining contact with patients before clinical examinations, allowing patients to compare the services of different doctors, thus

increasing patient satisfaction [7]. Therefore, it is of great practical significance to further explore the inner mechanism of doctor-patient information exchange in health communities both from a macroscopic perspective for the harmonious coexistence of doctor–patient relationships and for the effective utilization of medical resources from a microscopic perspective for the satisfaction of both doctors and patients in online health community information exchange.

Extensive research has been conducted on various antecedents that influence knowledge-sharing among online community users [8–11]. However, little research has been conducted to investigate the influence of different personality traits on the users' knowledge-sharing intentions in online communities from the perspective of their personality traits. At the root, personality traits drive individuals' complex behaviors; individual motivation and social capital are related to personality traits, and the effective transfer of knowledge through the behaviors of individuals with different personalities affects their judgment of whether to adopt knowledge-sharing [12]. Second, although we have learned a lot about the antecedents of users' knowledge-sharing intention in online health communities, these factors revolve around a single user's subjective feelings, such as a sense of self-worth, members' perceptions of social support and reputation enhancement [13], users' attitudes [14], and the lack of a systematic mechanistic framework. Finally, previous studies on knowledge-sharing in online health communities have focused on patients or physicians [15], and fewer studies have combined physicians and patients for comparative analysis. To fill the above research gaps, based on the personality trait theory, this paper explores the mechanisms of different personality traits on users' knowledge-sharing intentions in online health communities. At the same time, this paper considers the network relationship benefits of social capital, introduces social capital theory into the study of the mechanisms affecting the users' knowledge-sharing intention in the field of online health communities, constructs a theoretical model of how individuals with different personality traits in online health communities stimulate their knowledge-sharing intention through different social capital and try to elucidate how users with different personality traits stimulate their knowledge-sharing intention in the process of sharing information on online health communities in the structural dimension and relational dimension. The relationship between structural and relational capital on the knowledge-sharing intention of online health communities is examined. Finally, this paper examines the differences in the knowledge-sharing willingness between doctors and patients to bridge the gap between these two types of user differences.

Based on the personality trait theory and social capital theory, this paper establishes a new framework on the influence path of users' knowledge-sharing intention in online health communities, which takes the users' introversion-extraversion personality traits as independent variables, their knowledge-sharing intention as dependent variables, and structural dimensional capital-interaction, relational dimensional capital in social capital theory-trust, and reciprocity as mediating variables, to verify the influence of users with different personality traits on their knowledge-sharing willingness and the intermediate mechanism of action when opening the black box between users' different personality traits and their knowledge-sharing willingness in online health platforms. At the same time, this paper introduces social capital theory into the field of online health community information sharing and further validates the explanatory logic of the social capital theory. Therefore, this paper has important theoretical and practical significance for enriching the research on knowledge-sharing in online health communities, grasping the mechanisms of action and factors that affect the knowledge-sharing of online health community users, stimulating users' knowledge-sharing intentions, and helping to promote online health community operators to clarify their operation mechanisms.

This paper has the following innovative points: first, this paper introduces social capital theory into the field of online health community users' knowledge-sharing research, explains the factors affecting the online health community users' knowledge-sharing willingness from the perspective of social capital theory, and establishes a systematic mechanistic framework. Second, this paper integrates and compares two types of users (doctors

and patients) in online health communities and examines the differences in the knowledge-sharing willingness between doctors and patients with different personality traits, bridging the gap between the two types of users in online health communities, which is rarely studied in concert. Third, based on the personality trait theory, this paper considers the users of online health communities as individuals with both introversion-extraversion personality traits and explores the influence of different personality traits on knowledge-sharing willingness from the root, enriching the research on knowledge-sharing in online health communities.

## 2. Theoretical Analysis and Research Hypothesis

### 2.1. Personality Traits

Traits are perceptual contextual systems and tendencies that are unique to individuals in different contexts, are consistent and stable, and can govern an individual's behavior [16]. An introverted personality is one of the most important concepts in personality psychology and is closely related to knowledge-sharing and health [17]. Extraversion is a personality trait used to describe an individual's optimism, enthusiasm, and preference for socializing, and extraverted individuals have a high level of social competence [18]. An individual is said to be introverted if he or she believes that events occur as a result of personal abilities and attributes that can be determined or controlled by subjective, internal, personal actions [19], and introverted individuals are introverted, quiet, and reticent, preferring solitude or associating with only a few close friends [19]. It has been suggested that introversion-extraversion personality traits are important factors that influence a users' information-sharing behaviors in online communities [12]. This study focuses on introversion-extraversion personality traits because this study focuses on the communicative interactions of users in online health communities, and introversion-extraversion personality traits focus on reflecting the users' online social skills. The introversion-extraversion personality traits of online community users change as they experience them, and each individual has both introversion-extraversion personality traits. Thus, this paper argues that both dimensions may influence a users' social capital and knowledge-sharing intention, which is the theoretical basis for this paper's classification of the independent variables into two dimensions: introversion personality traits and extroversion personality traits.

### 2.2. Social Capital Theory

The unique social and interactive nature of online network communities dictates that social capital plays an important influence on individual knowledge-sharing [20]. The central claim of the social capital theory is that social networks constitute a valuable resource for social affairs, providing other members with "collectively owned capital" that they are entitled to use [21]. Social capital involves structural and relational dimensions. The structural dimension refers to the structure of the relationship between the two parties of the information interaction: specifically the interaction status of the two parties of the information interaction [22]. Structural dimensional capital provides members with access to relevant knowledge [23], and interactions can help to understand the network of relationships between individuals. Interactions refer to the structural dimensional links or bonds between individuals in a social network [24]. The interactions associated with networks can provide members with access to information and reduce the time and effort spent on gathering information [25]. The relational dimension of capital deals with the nature of connections between members in a community, and the key elements of this dimension are trust and reciprocity [25]. In online communities, trust is the subjective perception and expectation of individuals to respond to the knowledge of other members, providing a positive environmental climate for the whole community, and is a fundamental factor in knowledge-sharing among all parties [26].

### 2.3. Introversion-Extraversion Personality Traits and Knowledge-Sharing

Previous research has emphasized that extroverted individuals are more eager to share knowledge [27]. Extraverted individuals are more likely to interact with others and maintain positive connections [15]. In group-based activities, they are happy to participate in team discussions and collaborations, present their perspectives [27], and are more characterized by sharing knowledge with others. Enthusiasm and talkativeness form a physician's confidence in the counseling relationship based on the profession itself, and the physician's traits can influence the effectiveness of the counseling. Extraversion is a personality trait of physicians who can reflect the nature of their work and are more willing to share this health knowledge because of their friendly attitude and altruistic behavior. Introverted individuals are cautious about new things [12], but OHCs help patients improve their mental and behavioral health by designing customized needs for users [28], which increases patient satisfaction with receiving medical services. Therefore, patients with introverted personalities also want to gain the knowledge to address their illness in a healthy community.

Despite the differences between online health community settings and offline hospital settings, individual personalities and behavior in this network are still based on their experiences of reality. OHCs provide a venue where patients can discuss sensitive health issues (e.g., skin conditions, pregnancy) [29], and patients who feel inferior about their disease may not be active in knowledge-sharing and are more willing to listen to what others have to share. However, physicians have a strong sense of responsibility and compassion, and they provide health advice and expertise to their patients in a spirit of professional altruism [18]. Therefore, physicians will be more active in sharing their knowledge with the health community. Based on the above reasoning, the following hypothesis is proposed in this paper.

**H1a.** *Introversion personality traits positively affect the knowledge-sharing intention of physicians and patients.*

**H1b.** *Physicians' introversion personality traits are more positively correlated with knowledge-sharing intentions than patients.*

**H1c.** *Extraversion personality traits positively affect the knowledge-sharing intention of physicians and patients.*

**H1d.** *Physicians' extraversion personality traits are more positively correlated with knowledge-sharing intentions than patients.*

### 2.4. Introversion-Extraversion Personality Traits and Structural Dimensional Capital

Research on personality traits has found a positive association between personality traits and communication interactions. Extraversion implies energy for new things, and extroverted individuals are more likely to positively influence the activities of online communities [30]. Social involvement is the main characteristic of extroverted individuals [19], and physicians or patients with extraversion personality traits actively participate in community interactions, and friendly communication between both parties can effectively promote the sharing of health knowledge in online communities. Although introverted individuals tend to show poor socialization and are even in a marginal position in the social network, weak relationships are more likely to be a bridge for unfamiliar individuals than strong relationships in this online health community with a large spatial span and wide scope of involvement [31]. Patients with introversion are less likely to be noticed and trusted by others, but in an online health community with a large number of users who suffer from the same condition, they can also participate in communication and sharing with other patients to find solutions to their problems.

The success of online counseling depends on the actual use of the users [32]. Previous research has found that individuals ask for information from people they believe can provide helpful knowledge and who will not cause them to experience barriers to access in the

process [31]. The trait that extroverted individuals are good communicators can promote a sense of trust between the doctor and patient and enhance emotional connections during the communication process. Introverted individuals want to obtain the information they need from OHCs, and their purpose of use makes them participate in online health community interactions as well. Professional doctors are the knowledge providers in OHCs, and they want to disseminate more effective health information and treatment advice through communication with patients. Therefore, physicians are more likely to actively interact with others than patients. The following hypothesis is proposed in this paper.

**H2a.** *Introversion personality traits positively influence the interaction between physicians and patients.*

**H2b.** *Physicians' introversion personality traits are more positively correlated with interaction than patients.*

**H2c.** *Extraversion personality traits positively influence the interaction between physicians and patients.*

**H2d.** *Physicians' extraversion personality traits are more positively correlated with interaction than patients.*

### 2.5. Structural Dimensional Capital and Knowledge-Sharing

Interaction refers to interpersonal behaviors or relationships between individuals and reflects the level of time commitment in terms of the frequency of interaction among online community members [33]. OHCs provide a service for physicians and patients to share health knowledge, effectively promoting frequent social interactions between community members and others, even with people they have never met in real life [34], resulting in a broader range of information and experiences to share. As healthcare networks become more widespread, physicians and patients are increasingly interacting on social networks, providing opportunities for the widespread dissemination of public health information and other health knowledge [35]. The closeness of the interaction can influence members' positive attitudes toward sharing knowledge, and individual members can interact to gain access to others' information and resources and gain opportunities to exchange knowledge and expected value [36].

The network provides opportunities to combine and exchange knowledge, and interaction facilitates the exchange and sharing of group information and resources [24]. Patients want to communicate with their physicians to learn more about their health conditions and to seek emotional support. However, OHCs open some patients' information to the public, preventing the interactive behavior and knowledge-sharing intention of some patients who do not want others to see their privacy. Despite the lack of privacy in the online community setting [37], physicians will adhere to professional guidelines and will not readily divulge information about their patients, and will actively promote health knowledge in the community after learning more about the wide variety of patient conditions. Thus, physicians will have a higher knowledge-sharing intention due to the opportunity to access richer information. Therefore, the following hypothesis is proposed in this paper.

**H3a.** *Interaction positively influences the knowledge-sharing intention of physicians and patients.*

**H3b.** *Physicians' interactions have a stronger positive correlation with knowledge-sharing intentions than patients.*

### 2.6. The Mediating Role of Structural Dimensional Capital

Extroverted individuals will face their mental illness with optimistic emotions, and they will be more active in communication because they believe that timely and frequent interactions may reduce psychological stress when seeking treatment advice from doctors or communicating with other patients about their illnesses. Interaction is a channel for the flow of information and resources, and strong social networks and frequent social interactions increase members' knowledge-sharing intention [38]. Thus, interaction mediates the relationship between extraversion personality traits and knowledge-sharing.



Introverted individuals are more hesitant to share their thoughts when compared to extroverted individuals [39]. However, health-related personality factors might also help explain why certain groups have significantly better or worse health than others [40]. For example, introverted individuals at the margins of the network are influenced by individuals at the center, are exposed to some patients with similar health conditions, and have increased opportunities to interact and communicate with other patients. Individuals who have network relationships with other members develop more interactive relationships with others as they gain knowledge and resources, which makes it easier to transfer knowledge and information to other members [41]. Introverted individuals increase their willingness to share health knowledge through the "bridging" role of interaction. Thus, this paper argues that interaction mediates the relationship between introverted personality traits and knowledge-sharing. Based on the above reasoning, the following hypothesis is proposed in this paper.

**H4a.** *Interactions mediate the relationship between introversion personality traits and knowledge-sharing intentions.*

**H4b.** *Interactions mediate the relationship between extraversion personality traits and knowledge-sharing intentions.*

*2.7. Structural Dimensional Capital and Relationship Dimension Capital*

The relational dimension of capital deals with the nature of connections between members in a community, and the key elements of this dimension are trust and reciprocity [25]. In online communities, trust is the subjective perception and expectation of individuals to respond to the knowledge of other members, providing a positive environmental climate for the community as a whole, and is a fundamental factor in knowledge-sharing among all parties [26]. Reciprocity is a conditional benefit: when one party provides a certain resource to another party, the former expects a certain benefit for what they provide [42]. Previous research has shown that structural dimensional social capital affects relational dimensional capital [43]. The relational aspect of relational dimensional capital can be enhanced by interactions because interactions provide time, opportunities, and motivation to engage with others [44]. As community members interact more frequently over time, their trusting relationships become more concrete, and they are more likely to perceive each other as trustworthy [45]. At the same time, frequent social interactions result in members sharing more information with others, thus creating more reciprocal relationships [45]. The importance of reciprocity in Chinese social relationships is further exemplified by the saying: "If you throw me a peach, you will be rewarded with a pear".

Trust and reciprocity are, at the same time, the roots of online health community interactions; only giving without trust and reciprocity is transient and cannot sustain the relationships among community members. To communicate effectively, community users need to provide personally identifiable information to build the foundation of relationships with others [46]. In this respect, physicians show an advantage over patients who have professional credentials and are better able to promote mutual trust in the process of communication with patients. Some patient users prefer to go to the hospital after online diagnosis so that patients can target more disease causes, treatment recommendations, and prevention strategies [15]. Doctors are not eager to benefit from their patients, while patients prefer to derive more from their OHCs. Therefore, the following hypothesis is proposed in this paper.

**H5a.** *Interactions positively affect the trust of physicians and patients.*

**H5b.** *Physicians' interactions are more positively correlated with trust than patients.*

**H5c.** *Interactions positively affect the reciprocity between physicians and patients.*

**H5d.** *Patients' interactions are more positively correlated with reciprocity than physicians.*

*2.8. Relationship Dimension Capital and Knowledge-Sharing*

Relational social capital has an emotionally connected nature that facilitates knowledge exchange between individuals [25], and the norms of trust and reciprocity may influence knowledge sharing [43]. It has been shown that informational support is the most common type of patient expectation in online healthcare communities; however, emotional support plays a more important role in helping patients move into a healthier state [47]. OHC allows users to exchange information with like-minded individuals with the same health condition and similar experiences for social support via the Internet [48,49]. Members' perceptions of social support and the higher reputation of physicians can positively influence online health community knowledge sharing through a sense of trust [13]. Trust allows individuals to rationalize their decisions and provides useful information for communication [43]. Physician-to-patient trust may play an important role in sharing solutions to pain [50]. Patients can receive emotional and informational support from online health communities [47], and to meet their information support desires, patients choose doctors they trust to provide them with information support and high professional capital [51] and are, therefore, more willing to share their conditions with them. Physicians' trust in the platform may reduce uncertainty about the system and related processes and better provide health information and consultation services to patients. Conversely, in the area of online healthcare services, trust issues can increase uncertainty about the system and related processes as patients cannot physically see the medical outcomes of other patients. To improve the efficiency and effectiveness of treatment, ordinary patients need not only to trust the community and create an atmosphere of knowledge sharing [25] but also to be informed of the physician's treatment ability, believe that the physician will put the patient's interests first [52], actively communicate with the physician about their condition, and increase their willingness to share knowledge with the psychological community. Thus, individuals with a higher propensity to trust may have a higher knowledge-sharing intention.

Social interactions and relationship building are key to providing information exchange and conducting social support [53]. Reciprocity promotes the knowledge-sharing intentions of online community users [54]. Knowledge source credibility and positive emotions influence individuals' acceptance and the sharing of health knowledge [55]. Unlike many online communities, the primary purpose of online healthcare users is not monetary rewards, as they are willing to establish long-term reciprocal relationships with other members to ensure the continued development of knowledge-sharing [15]. Members have a greater incentive to contribute knowledge if the input of knowledge-sharing is rewarded, and strong reciprocal relationships promote knowledge-sharing intentions among members [24]. Specifically, when reciprocity norms exist in a community, individuals believe that their knowledge contributions will be rewarded, thus ensuring continued contributions [20].

In OHCs, because patients lack expertise related to treatment and physicians are unfamiliar with the patient's conditions, they decide whether to trust or reward each other by evaluating the information obtained during the healthcare service, which affects knowledge-sharing among community members. Whether patients are distracted by misinformation when seeking healthcare services through online communities is a matter of considerable concern, and information from unreliable sources may confuse information seekers [56]. Patient trust in the online environment can be disturbed by these factors, reducing overall trust in online healthcare services. The purpose of patient participation in OHCs is to gain health knowledge and emotional support [5], where the medical diagnosis reports obtained after communication with a physician are a reflection of the patient's need for reciprocity. These factors enhance patients' reciprocal expectations of physicians and promote the dissemination and sharing of health knowledge by patients. Based on the above reasoning, the following hypothesis is proposed in this paper.

**H6a.** *Trust positively affects the physicians' and patients' knowledge-sharing intention.*

**H6b.** *Physicians' trust is more positively correlated with knowledge-sharing intentions than patients.*

**H6c.** *Reciprocity positively affects the physicians' and patients' knowledge-sharing intention.*

**H6d.** *Patients' reciprocity is more positively correlated with knowledge-sharing intentions than physicians.*

*2.9. The Mediating Role of Relationship Dimension Capital*

Trust among members is driven by high levels of interaction, and individuals with a higher frequency of interaction with others may trust other members more [57]. Over time, interactions between members become more specific, and individuals who occupy a central position in a network of social interactions may be perceived as trustworthy by other members of the network [23]. Social interactions have the function of conveying health information and seeking advice on medical conditions and can provide useful resources for both physicians and patients, making the trust relationship stronger. Trust establishes and maintains the interaction between members, which increases their willingness to share quality knowledge [24]. Thus, trust mediates the role between interaction and knowledge-sharing.

Frequent social interactions lead individuals to share more information with other members, and the information contributions of others meet individuals' expectations, creating more reciprocal relationships [45]. The anonymous nature of online communities leads to a decrease in the perceptions of individual differences and an increase in adherence to group norms, enhancing the interactive behavior of members to participate in discussions and making individuals more pro-social and reciprocal [58]. In OHCs, member interaction creates a sense of mutual benefit, and reciprocity is a relative and fair code of conduct for exchanging information and knowledge, and this reciprocity can drive members' knowledge-sharing intentions when individuals develop the idea of mutual benefit with other members [20]. Thus, reciprocity mediates the role between interaction and knowledge-sharing. Therefore, the following hypothesis is proposed in this paper.

**H7a.** *Trust mediates the relationship between interaction and knowledge-sharing intentions.*

**H7b.** *Reciprocity mediates the relationship between interaction and knowledge-sharing intentions.*

Through a systematic review of the existing literature, this paper concludes that although the theoretical exploration of online health communities has attracted the attention of many scholars, there are still shortcomings. For example, the literature has paid little attention to the mechanism of personality traits affecting users' knowledge-sharing intentions in online health communities, and there is a lack of comparative studies on the different users of online health communities. Therefore, this paper introduces social capital theory into the study of users' knowledge-sharing intentions in online health communities and explores the intermediate mechanisms of action between users' traits and their knowledge-sharing intentions in online health community scenarios, as well as the differences between patients and doctors in this influence relationship. Based on the above theoretical analysis, this study constructs a theoretical model, as shown in Figure 1. The model uses introversion-extraversion personality traits as independent variables, knowledge-sharing intentions as dependent variables, and interaction, trust, and reciprocity as mediating variables to examine the main effects and the influence of introversion personality traits on knowledge-sharing intentions and the chain mediating role of interaction, trust, and reciprocity while comparing and analyzing the differences between doctors and patients in this process.

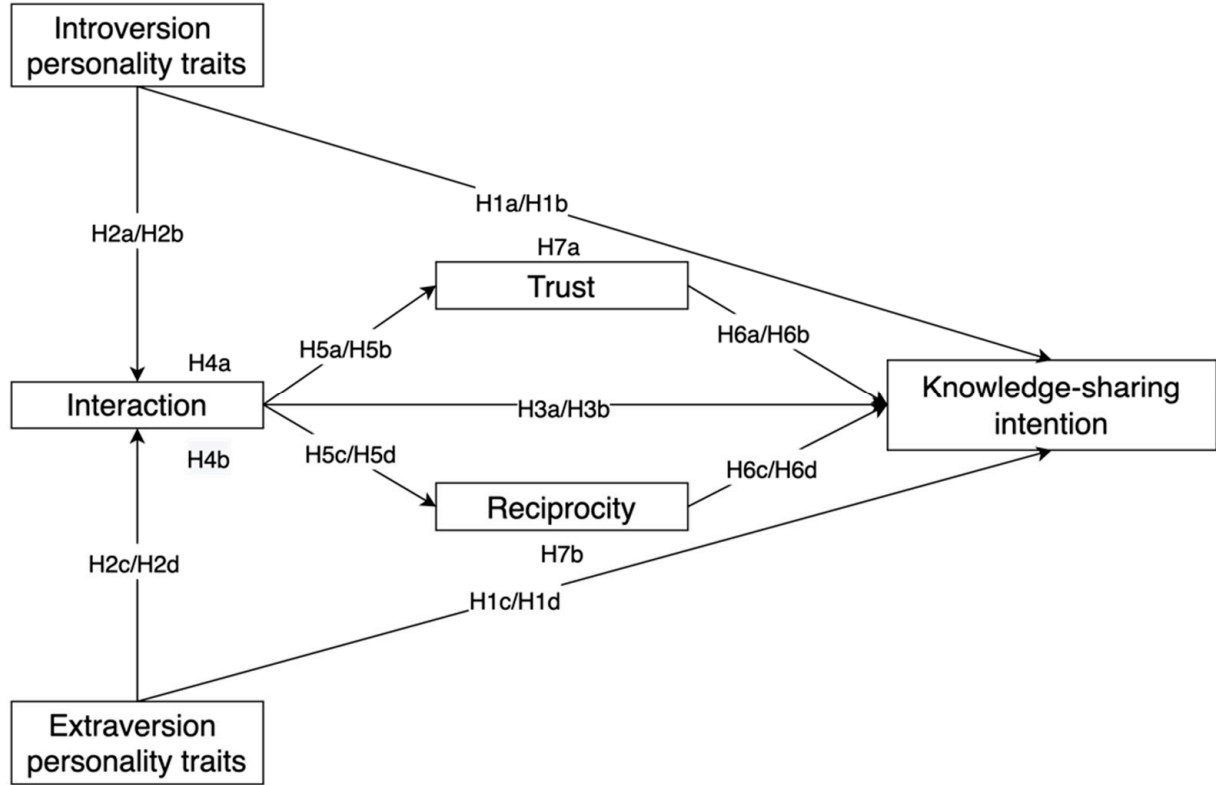

**Figure 1.** Research model.

### 3. Methodology and Measurement

#### 3.1. Sampling Strategy and Sample Collection

This study used a random sampling method to collect primary data by distributing questionnaires from three well-known Chinese online health communities, 39Health.com (accessed on 20 November 2022), Dingxiangyuan, and Hodafu. The questionnaire was divided into two parts: a section on demographic characteristics, including gender, age, education level, length of use, and income, and the main part of the questionnaire.

To ensure the validity and reliability of the questionnaire, two professors in the field of knowledge management and three senior managers of the online health community were invited to remove some ambiguous items. The revised questionnaire was pre-tested among 30 members of the online health community, and the question items were again further revised based on the results. Finally, a formal questionnaire was determined, consisting of 21 questions. We were able to distinguish between health practitioners and patients by investigating occupational and educational backgrounds in the questionnaire. Respondents were classified as doctors if they had experience working in the medical and health care field or had professional medical and health care education otherwise, respondents were classified as patients.

In this study, 703 questionnaires were distributed, 593 questionnaires were returned, and the questionnaires were screened to eliminate 22 questionnaires in which all answers were the same and incomplete, and finally, 571 valid questionnaires were obtained. The results showed that 243 of the 571 valid respondents were doctors and 328 respondents were patients, with 56.74% males ($n$ = 324) and 43.26% females ($n$ = 247). This paper used SPSS22.0 software to analyze the sample data, and the descriptive statistics of the sample are shown in Table 1.

**Table 1.** Descriptive statistics.

| Category | | Total (*n* = 571)) | Doctor (*n* = 243) | Patient (*n* = 328) |
|---|---|---|---|---|
| Sex | Man | 324 (56.74%) | 160 (48.78%) | 164 (67.49%) |
| | Woman | 247 (43.25%) | 168 (51.22%) | 79 (32.51%) |
| Age | <25 | 46 (8.056%) | 31 (9.451%) | 15 (6.173%) |
| | 25~30 | 149 (26.095%) | 101 (30.793%) | 48 (19.753%) |
| | 31~40 | 205 (35.902%) | 107 (32.622%) | 98 (40.329%) |
| | 41~50 | 118 (20.665%) | 55 (16.768%) | 63 (25.926%) |
| | >50 | 53 (9.282%) | 34 (10.366%) | 19 (7.819%) |
| Education level | Below high school | 13 (2.277%) | 7 (2.134%) | 6 (2.469%) |
| | College | 288 (50.438%) | 172 (52.439%) | 116 (47.737%) |
| | Master and above | 166 (29.072%) | 96 (29.268%) | 70 (28.807%) |
| Monthly income | <2001 | 65 (11.384%) | 30 (9.146%) | 35 (14.403%) |
| | 2001~4000 | 28 (4.904%) | 17 (5.183%) | 11 (4.527%) |
| | 4001~6000 | 11 (1.926%) | 6 (1.829%) | 5 (2.058%) |
| | 6001~8000 | 35 (6.13%) | 35 (10.671%) | 0 (0%) |
| | 8001~10,000 | 352 (61.646%) | 213 (64.939%) | 139 (57.202%) |
| | >10,000 | 184 (32.224%) | 80 (24.39%) | 104 (42.798%) |
| Involving time | <6 months | 33 (5.779%) | 20 (6.098%) | 13 (5.35%) |
| | 6~12 months | 38 (6.655%) | 22 (6.707%) | 16 (6.584%) |
| | 12~24 months | 53 (9.282%) | 34 (10.366%) | 19 (7.819%) |
| | 24~36 months | 110 (19.264%) | 75 (22.866%) | 35 (14.403%) |
| | >36 months | 337 (59.019%) | 177 (53.963%) | 160 (65.844%) |

*3.2. Measurement of Constructs*

To ensure reliability, all questions in this paper were measured using well-established national and international scales. Each item is measured on a 7-point Likert scale, where 1 means 'strongly disagree' and 7 means 'strongly agree'.

Introversion–extraversion personality traits (INT and EXT). This construct measures the extent to which the users of online health communities are motivated to engage in knowledge-sharing [45]. This study adapted Saucier's construct measure [59]. There are four questions for INT and four for EXT, with a total of eight questions.

Interaction (INA). This construct is considered to be a channel for information and resource retention; online community interactions encompass the strength of relationships between the members, time spent, and frequency of communication, and interactive learning allows for the acquisition of observable knowledge [20]. In this study, three question items were set to measure INA, as suggested by C. M. Chiu [24].

Trust (TRU). Trust is the expectation of individual members that other members will adhere to guidelines and norms, and members in a social network will be less suspicious of other members taking advantage of them in a trusting relationship [46]. In this study, four questions were developed to measure trust between doctors and patients in the health community, which were drawn from the study by Chang et al. [25].

Reciprocity (REC). Reciprocity is considered to be a positive behavior by both parties adhering to the principle of fairness when members expect that others will provide the help they expect [41]. In this study, three question items were set for this variable based on Wasko's suggestion [20].

Knowledge-sharing intention (KSI). Online community knowledge-sharing is the contribution of individual members' knowledge to the community [50], that is, assisting others to develop the capacity for effective action. In this paper, three questions were set for KSI based on Huang's research [60].

Specific Measurement scales and construct items are shown in Table A1 of Appendix A.

## 4. Data Analysis

### 4.1. Reliability and Validity

The data analysis followed a two-stage approach: a measurement model and a structural model. First, the validity and reliability of the six elements of the measurement model (i.e., inside-out personality traits, interaction, trust, reciprocity, and knowledge-sharing intention) were assessed, followed by a validating factor analysis through structural equations, which focused on examining the structural relationships between underlying variables.

Based on Fornell et al.'s two-step approach, this study examined the internal validity and reliability of the constructs measuring the model [61]. First, the internal consistency of the constructs was assessed using two measures. Table 2 shows the results for the validated factor analysis of the measurement model, with Cronbach's alpha for each of the resulting constructs appearing above 0.796, exceeding the recommended 0.7. Therefore, the reliability of the constructs in this study was good.

**Table 2.** Reliability and validity.

| Items | Doctors | | | Patients | | |
| --- | --- | --- | --- | --- | --- | --- |
| | Cronbach's Alpha | AVE | CR | Cronbach's Alpha | AVE | CR |
| INT | 0.893 | 0.683 | 0.896 | 0.889 | 0.678 | 0.893 |
| EXT | 0.869 | 0.631 | 0.872 | 0.895 | 0.685 | 0.897 |
| INA | 0.855 | 0.666 | 0.857 | 0.817 | 0.613 | 0.824 |
| TRU | 0.865 | 0.626 | 0.87 | 0.889 | 0.676 | 0.893 |
| REC | 0.822 | 0.62 | 0.83 | 0.864 | 0.69 | 0.87 |
| KSI | 0.796 | 0.571 | 0.8 | 0.827 | 0.617 | 0.828 |

Notes: INT—introversion personality traits; EXT—extraversion personality traits; INA—interaction; TRU—trust; REC—reciprocity; KSI—knowledge-sharing intention.

Second, the convergent validity and discriminant validity of the constructs were tested using two measures. Table 2 shows that the AVEs of the measured variables were all exceedingly higher than 0.571, which exceeded the recommended value of 0.5. Therefore, this study has good convergent validity. We verified the discriminant validity by testing whether the correlation between the constructs was less than the square root of the AVE. In Table 3, the main diagonal value is the square root of the AVE, and the non-main diagonal is the correlation coefficient between the constructs, with all diagonal values exceeding the correlation between any pair of constructs. This result indicates that the measurement model also had sufficient discriminant validity.

**Table 3.** Descriptive statistical analysis results.

| | Mean | SD | INT | EXT | INA | TRU | REC | KSI |
| --- | --- | --- | --- | --- | --- | --- | --- | --- |
| | | | | | Doctors | | | |
| INT | 4.383 | 1.507 | 0.826 | | | | | |
| EXT | 4.366 | 1.568 | 0.314 *** | 0.794 | | | | |
| INA | 4.535 | 1.636 | 0.479 *** | 0.500 *** | 0.816 | | | |
| TRU | 5.062 | 1.464 | 0.335 *** | 0.305 *** | 0.491 *** | 0.791 | | |
| REC | 4.680 | 1.602 | 0.322 *** | 0.290 *** | 0.421 *** | 0.288 *** | 0.787 | |
| KSI | 4.930 | 1.222 | 0.610 *** | 0.560 *** | 0.692 *** | 0.599 *** | 0.591 *** | 0.756 |
| | | | | | Patients | | | |
| INT | 4.287 | 1.526 | 0.823 | | | | | |
| EXT | 4.740 | 1.636 | 0.268 *** | 0.828 | | | | |
| INA | 4.977 | 1.518 | 0.098 | 0.145 ** | 0.783 | | | |
| TRU | 5.152 | 1.468 | 0.078 | 0.071 | 0.147 ** | 0.822 | | |
| REC | 4.998 | 1.642 | 0.102 | 0.082 | 0.151 ** | 0.112 * | 0.831 | |
| KSI | 5.296 | 1.218 | 0.141 * | 0.328 *** | 0.330 *** | 0.146 ** | 0.182 *** | 0.785 |

Notes: *** $p < 0.001$. ** $p < 0.01$. * $p < 0.05$.

*4.2. Confirmatory Factor Analysis*

To further test the discriminant validity of the variable measures, confirmatory factor analysis was conducted using AMOS 22 software for INT, EXT, INA, TRU, REC, and KSI. The fit of the six-factor model for the physician group ($\chi^2$/df = 1.338, RMSEA = 0.037, GFI = 0.917, IFI = 0.980, CFI = 0.980, NFI = 0.926) was significantly better than the other models, and the fit of the six-factor model for the patient group ($\chi^2$/df = 1.263, RMSEA = 0.028, GFI = 0.942, IFI = 0.988, CFI = 0.987, NFI = 0.943) was also significantly better than the other models, which suggests that the variables have good discriminant validity.

*4.3. Common Method Variance Test*

To avoid the problem of common method bias from influencing the findings of this study, a common method bias was controlled and tested in terms of both procedural and statistical methods. In terms of procedural design, the study draws on the following measures: developing clear and concise questions, using anonymous questionnaires for collection, and varying how variables are obtained and measured to minimize respondent guesswork about the purpose of the measurement. For statistical testing, this study utilized Harman's one-way test to verify the extent of the homogeneity error, which is an exploratory factor analysis of the full set of constructs, and if the variance explained by the first factor exceeded 50%, this indicated a high common method bias in the data. The results of this study, calculated using SPSS 22.0, showed that the variance explained by the first factor in the healthy physician group was 40.05%, and the variance explained by the first factor in the patient group was 23.07%, with both less than 50 %. Therefore, there was no serious problem of common method bias in the results of this study.

*4.4. Hypotheses Testing and Multi-Group Analysis*

4.4.1. Direct Effects Test

In this paper, we used AMOS to test the path coefficients of the hypothesized model for the effect of the introversion-extraversion personality traits of doctors and patients on their knowledge-sharing intention, and the results are shown in Table 4. Among them, the positive effects of introversion personality traits on interactions ($\beta$ = 0.089, *p* = 0.174) and knowledge-sharing intentions ($\beta$ = 0.004, *p* = 0.952) were not significant, and the positive effect of trust on knowledge-sharing intentions ($\beta$ = 0.065, *p* = 0.274) was not significant in the patient group, while the rest of the paths were significant. Thus H1c, H2c, H3a, H5a, H5c, and H6c are supported and H1a, H2a, and H6a are partially supported (physician part).

**Table 4.** Multi-group path analysis and difference comparison.

| Regression Path | Doctors | | | Patients | | | Doctors-Patients Path Differences Comparison | | |
|---|---|---|---|---|---|---|---|---|---|
| | Path Co-efficient | *p* | Result | Path Co-efficient | *p* | Result | Differences | *p* | Result |
| INA←INT | 0.403 | *** | H2a (S) | 0.089 | 0.174 | H2a (NS) | 0.314 | 0.003 | H2b (S) |
| INA←EXT | 0.439 | *** | H2c (S) | 0.147 | 0.026 | H2c (S) | 0.292 | *** | H2d (S) |
| TRU←INA | 0.581 | *** | H5a (S) | 0.185 | 0.004 | H5a (S) | 0.396 | *** | H5b (S) |
| REC←INA | 0.526 | *** | H5c (S) | 0.164 | 0.012 | H5c (S) | 0.362 | *** | H5d (S) |
| KSI←INT | 0.304 | *** | H1a (S) | 0.004 | 0.952 | H1a (NS) | 0.3 | *** | H1b (S) |
| KSI←EXT | 0.209 | *** | H1c (S) | 0.305 | *** | H1c (S) | −0.096 | 0.159 | H1d (NS) |
| KSI←INA | 0.205 | 0.009 | H3a (S) | 0.308 | *** | H3a (S) | −0.103 | 0.138 | H3b (NS) |
| KSI←TRU | 0.31 | *** | H6a (S) | 0.065 | 0.274 | H6a (NS) | 0.245 | 0.002 | H6b (S) |
| KSI←REC | 0.339 | *** | H6c (S) | 0.137 | 0.023 | H6c (S) | 0.202 | 0.009 | H6d (S) |

Notes: *** *p* < 0.001; S: support; NS: no support.

4.4.2. Indirect Effect Test

To further analyze the mediating role of interaction, trust, and reciprocity between the introversion-extraversion personality traits and knowledge-sharing intention, this paper

used the Bootstrap method to test the mediating effect, and if the 95% confidence interval did not contain zero, the results were statistically significant, indicating the existence of the mediating effect. The results are shown in Table 5. The results of the mediating effect test, both in the patient group and the physician group, showed that the mediation effects of the interaction between extraversion personality traits and knowledge-sharing intention and the mediation effects of reciprocity between interaction and knowledge-sharing intentions did not include zero. Therefore, interactions mediate the relationship between extraversion personality traits and knowledge-sharing intentions, and reciprocity mediates the relationship between interaction and knowledge-sharing intentions. H4b and H7b are supported. In the physician group, the mediation effects of the interaction between introversion personality traits and knowledge-sharing intentions and the mediation effects of trust between interaction and knowledge-sharing intentions did not include zero. However, in the patient group, the mediation effects of the interaction between introversion personality traits and knowledge-sharing intention and the mediation effects of trust between interaction and knowledge-sharing intention included zero. Therefore, interactions mediated between extraversion personality traits and knowledge-sharing intentions, and trust mediated between interactions and knowledge-sharing intentions in the physician group, whereas interactions did not mediate between extraversion personality traits and knowledge-sharing intentions, and trust did not mediate between interactions and knowledge-sharing intentions in the patient group; therefore, H4a and H7a were partially supported (physician part).

**Table 5.** Mediation test.

| | Regression Path | Effect | SE | Bias Corrected (95%) | | | Percentile Method (95%) | | |
|---|---|---|---|---|---|---|---|---|---|
| | | | | LLCI | ULCI | $p$ | LLCI | ULCI | $p$ |
| Doctors | INT→INA→KSI | 0.083 | 0.039 | 0.016 | 0.169 | 0.017 | 0.012 | 0.164 | 0.022 |
| | EXT→INA→KSI | 0.09 | 0.04 | 0.018 | 0.179 | 0.016 | 0.014 | 0.173 | 0.022 |
| | INA→TRU→KSI | 0.18 | 0.037 | 0.116 | 0.263 | 0 | 0.112 | 0.259 | 0 |
| | INA→REC→KSI | 0.178 | 0.037 | 0.115 | 0.259 | 0 | 0.112 | 0.256 | 0 |
| Patients | INT→INA→KSI | 0.028 | 0.022 | −0.008 | 0.08 | 0.131 | −0.01 | 0.075 | 0.171 |
| | EXT→INA→KSI | 0.045 | 0.022 | 0.01 | 0.096 | 0.014 | 0.007 | 0.092 | 0.022 |
| | INA→TRU→KSI | 0.012 | 0.013 | −0.006 | 0.048 | 0.198 | −0.009 | 0.043 | 0.288 |
| | INA→REC→KSI | 0.022 | 0.013 | 0.003 | 0.059 | 0.016 | 0.001 | 0.052 | 0.037 |

### 4.4.3. Multi-Group Analysis

To compare the pathway differences between physicians and patients, a multi-group structural equation model analysis using AMOS was conducted in this paper. The results of the multi-group analysis showed (see Table 3) that the standardized path coefficient differences between physicians' and patients' introversion personality traits and interaction on knowledge-sharing intentions were −0.096 and −0.103, respectively, with $p$-values of 0.159 and 0.138. It indicated that the differences between the physicians' and patients' introversion personality traits and interaction of knowledge-sharing intentions were not significant, and the differences in the remaining seven direct effects were all significant. H1b, H2b, H2d, H5b, H5d, H6b, and H6d were supported and H1d and H3b failed to be supported.

## 5. Discussion

The novel coronavirus epidemic has caused a global shortage of offline medical resources, and people are increasingly inclined to pre-search health information from the Internet. Research related to online health communities has attracted the attention of many scholars. However, systematic research on users' knowledge-sharing willingness in health communities and comparative research on different users are still inadequate, so this paper uses well-known Chinese online health communities as a data source, takes the knowledge-sharing intention of different users in online health communities as a research object, and analyzes the proposed hypothesis model. The results are as follows.

### 5.1. Theoretical Implications

From the perspectives of personality trait theory and social capital theory, this study expands the research on knowledge-sharing in OHCs by comparing the knowledge-sharing paths of two types of key users [47]. From the perspective of the introverted personality traits of users in OHCs, this paper examines the issues of personality traits and knowledge-sharing in OHCs that have hardly been investigated in previous studies [13,62], enriching the understanding of knowledge-sharing.

This study uses social capital theory to explain how doctors and patients with two personality traits can better stimulate their knowledge-sharing in OHCs. Interaction and reciprocity are the driving factors that explain users' participation in knowledge-sharing in online communities [54], and trust is a key factor that causes patients to reduce their knowledge-sharing intention [63]. Social capital is the mechanism that facilitates individual collaboration in OHCs: a concept that explains an individual's potential or real capital with friends or strangers [20]. Members in a social network can obtain different resources according to their positions in the social relationship structure and can also invest in other resources, expecting to obtain future benefits [21]. The social capital theory can also explain the expectations and reality of users in the health community [36]. Patients acquire health knowledge and share experiences in communication with doctors or other patients. Doctors guide patients in scientific treatment and cultivate doctor–patient relationships [15,24]. The desire to continue to profit in the community has stimulated the knowledge-sharing intention of both types of users.

This paper develops a study on knowledge-sharing between physicians and patients by building a model of comparative mechanisms. The study found that doctors with introverted personality traits were more willing to share knowledge than patients, and doctors with both introversion and extraversion personality traits were more willing to participate in community interactions. For doctors, interactions can generate more trust than patients [57], and further, trust can stimulate more knowledge-sharing than patients [46,50,56]. Additionally, for patients, more reciprocity results in more knowledge-sharing than for doctors.

### 5.2. Practical Implications

OHCs should divide their users into doctors and patients and should conduct differentiated management according to the interests and needs of the two parties.

OHCs should set incentives for active users, especially for patients. To encourage highly introverted patients to participate in interactions and to share their doubts and experiences, OHCs can use the platform database to observe the number and content of posts, can frequently ask patients to provide more information on healthcare or disease prevention, and can provide patients with emotional support to create an atmosphere of emotional exchange.

OHCs should create new sections to provide interaction and communication opportunities for introverted patients and establish a mutually beneficial mechanism that is more focused on the needs of patients. Meanwhile, OHCs should reward doctors for their attitude and behavior.

OHCs could establish reputation-scoring systems that allow patients to evaluate services after seeking treatment. Then, the OHCs could archive and store electronic medical records, establishing a database for patients to prompt patients to undergo their next treatment, thus enhancing the patient's trust in the platform and promoting the sustainable development of OHCs.

OHCs should also establish trust and reciprocal emotional channels to create a warm emotional atmosphere. In such an atmosphere, patients are more likely to communicate more with doctors to obtain more targeted treatment. Meanwhile, OHCs should establish a mutually beneficial mechanism. When providing services, it should be more targeted to the needs of patients and enhance patients' benefits and rewards in many respects.

*5.3. Limitations and Future Research*

This study has some limitations. First, the sample size is not large. This study only collected data from three OHCs, and the sample size has not been able to represent all users of the OHCs. In the later stage, the sample size needs to be increased to further expand the collection scope. Second, this study only investigated introversion-extraversion personality traits, and whether other personalities affect the knowledge-sharing of these two types of users in the OHCs still needs to be explored. Third, this study only incorporated the structural dimension and relational dimension of social capital into the model and did not examine the role of the cognitive dimension. Future research should consider the impact of the cognitive dimension of capital. Fourth, this study uses interactions to represent the structural dimension and trust and reciprocity to represent the relational dimension. However, there are other factors, such as network centrality, comments, etc., that can better enrich the explanation of each dimension of social capital.

## 6. Conclusions

This study explains the mechanisms by which both physicians' and patients' introversion-extraversion personality traits influence their knowledge-sharing intentions. First, the effect of patients' introverted personality traits on knowledge-sharing intentions and interactions is insignificant, while physicians' introverted personality traits positively influence their interaction and knowledge-sharing intention in online health communities. Meanwhile, compared to patients, doctors' introversion personality traits positively influence interaction and knowledge-sharing intentions more strongly, which is related to doctors' professional ethics and emphasis on professional altruism.

Second, extraversion personality traits positively influence both physicians' and patients' interaction and knowledge-sharing intentions in online health communities. Additionally, compared to patients, the positive effect of extraversion personality traits on interactions is stronger for physicians, but the difference in the effect on the knowledge-sharing intention was not significant.

Third, interaction in online health communities positively affects doctor-patient trust and reciprocity. The more interaction there is between doctors and patients and between patients and patients, the closer the relationship is and the more likely it is to generate a sense of trust and reciprocity. Meanwhile, physician interaction and reciprocity positively affect their knowledge-sharing intention. Patients' sense of reciprocity positively influences their knowledge-sharing intention, while the effect of interaction on their knowledge-sharing intention is not significant. In addition, among these four path relationships, physicians have a stronger influence relationship compared to patients, which is related to the dominance of physicians in the knowledge sharing of online health communities.

Fourth, interactions mediate between extraversion personality traits and knowledge-sharing intentions, and reciprocity mediates between interaction and knowledge-sharing intentions. This suggests that extroverted doctors and patients are more willing to share knowledge and information by communicating with others on online health platforms, and that information sharing between doctors and patients is based on the premise of mutual benefit and reciprocity.

**Author Contributions:** Conceptualization, M.L. and Y.S.; methodology, Y.S.; software, B.S.; validation, M.L., Y.S. and B.S.; formal analysis, B.S. and M.L.; investigation, M.L.; resources, Y.S.; data curation, writing—original draft preparation B.S.; writing—review and editing, funding acquisition, Y.S. All authors have read and agreed to the published version of the manuscript.

**Funding:** This research was funded by the Key Project of Social Science Planning of the Beijing Municipal Education Commission, grant number SZ202010011007.

**Institutional Review Board Statement:** Not applicable.

**Informed Consent Statement:** Not applicable.

**Data Availability Statement:** Not applicable.

**Conflicts of Interest:** The authors declare no conflict of interest.

**Appendix A**

**Table A1.** Measurement scales and construct items.

| Measurement Items | Factor Loading | |
|---|---|---|
| | DO | PA |
| Introversion (INT) | | |
| INT1: Shy | 0.792 | 0.727 |
| INT2: Quiet | 0.779 | 0.784 |
| INT3: Bashful | 0.866 | 0.888 |
| INT4: Withdrawn | 0.865 | 0.883 |
| Extraversion (EXT) | | |
| EXT1: Talkative | 0.747 | 0.804 |
| EXT2: Extroverted | 0.862 | 0.876 |
| EXT3: Bold | 0.789 | 0.824 |
| EXT4: Energetic | 0.775 | 0.804 |
| Interaction (INA) | | |
| INA1: I spend a lot of time interacting with some members in this OHC | 0.792 | 0.886 |
| INA1: I have frequent communication with some members in this OHC | 0.789 | 0.697 |
| INA1: I maintain close social relationships with some members in this OHC | 0.865 | 0.753 |
| Trust (TRU) | | |
| TRU1: Members in this OHCs will not take advantage of others even when the opportunity arises | 0.803 | 0.829 |
| TRU2: Members in this OHC will always keep the promises they make to one another | 0.805 | 0.841 |
| TRU3: Members in this OHC would not knowingly do anything to disrupt the conversation | 0.751 | 0.806 |
| TRU4: Members in this OHC are truthful in dealing with one another | 0.805 | 0.813 |
| Reciprocity (REC) | | |
| REC1: When I share knowledge in this OHC, I believe that my questions will be answered in the future | 0.827 | 0.855 |
| REC2: I believe that other members I interact with would help me if I was in need | 0.759 | 0.812 |
| REC3: When I share my knowledge in this OHC, I expect some other members to respond when I am in need | 0.774 | 0.825 |
| Knowledge-sharing (KSI) | | |
| KSI1: I intend to share knowledge with other members | 0.716 | 0.750 |
| KSI2: I am always willing to share knowledge with other members when they ask | 0.770 | 0.819 |
| KSI3: I am always trying to share knowledge with other members | 0.780 | 0.786 |

Note: DO—doctors; PA—patients.

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
