# Peer review of "Impact of Introversion-Extraversion Personality Traits on Knowledge-Sharing Intention in Online Health Communities: A Multi-Group Analysis"

_sustainability, doi:10.3390/su15010417_

Round 1
Reviewer 1 Report
This manuscript investigated two types of users in online health communities and analyzed the knowledge-sharing paths of the two types of users. However, the reviewer has some major concerns as follows:
1) The introduction and literature review need to be improved. The knowledge-sharing in online health communities is a hot topic in recent year. The research gap must be highlighted compared with the existing works. So that please add a new section before section 2.
2) The authors applied the two types from personality trait. The reason should be mentioned that why other three did not involved.
3) For the analysis tool, it would be better to use the Structural equation model software, e.g., amos. In this way, other test can be performed in addition to the current step. The more analysis about the proposed model must be added.
4) The discussion part should be separated into two part, section 5 discussion and section 6 conclusion. Currently, the discussion related to the results are not sufficient.
Reviewer 2 Report
General remarks
The manuscript aims to analyze how some personality traits, namely the degree of introversion/extroversion, may influence the intentions of sharing knowledge in online health communities. For this, the theories of social capital and personality traits are used. As for the data, they were obtained through the application of questionnaires in some well-known Chinese online health communities.
Specific remarks
I begin by acknowledging that I really enjoyed reading the manuscript. In fact, the subject under consideration is quite interesting, moderately relevant – one should recognize that this type of (online health) community, unfortunately, does not exist in many countries – and clearly appropriate for the section of the Journal to which it was submitted.
The theories of personality traits and social capital were well chosen as theoretical support. I must even say that, unfortunately, these seem not to have been taken into account in certain public health matters where some decisions are/were made without taking into account that, in fact, “one size does not fit all”.
As for the data processing methodology, it also seems to me that it was applied competently, and its results were well explored.
As for the data, I have, however, a doubt, which I ask the authors to clarify. Reading Table 1, on page 8, I see that 9 of the doctors have an education level below high school. If this interpretation is correct, how is this possible?

Reviewer 3 Report
1. According to figure 1, the title and the hypothesized theoretical model did not have consistency. I suggest the author should revise the title to meet the research questions and aims.
2. According to the section of the introduction, I suggest the author should employ the analysis of multiple groups to present and explore the relationships between the introversion-extraversion personality traits on their online community behaviors.
3. In addition, I suggest the author should consider the doctors and patients as multiple groups to present and explore their online community behaviors.
4. In section 2.8, I consider this issue is the core of this research. I suggest the author should focus on this concern to deal with the relationships among social capital, interaction, and online community through the literature analysis or theoretical discourse analysis.
5. In the section of the results, I suggest the author should follow my suggestions to revise the statistical analysis and provide robust results to meet the criteria of SEM.
6. Moreover, I suggest the author employ multiple-group analysis to deal with the groups of introversion-extraversion personality traits and groups of doctors and patients.
7. In table 4 and table 5, I suggest the author should employ SEM to deal with sample data and present the results of structural analysis and multiple-group analysis via SEM.
Round 2
Reviewer 1 Report
The reviewers' concern had been addressed. Please consider to accept the current version.
Reviewer 3 Report
The author made more revisions to this manuscript based on my concerns. I suggest this manuscript should be accepted in the present edition.